# The Foundational Components of Self-Regulating (Sustainable) Economies and Ecosystems: Implications for Green Infrastructure and Economic Restoration

John H. Giordanengo

Economic Restoration Services, Fort Collins, CO 80521, USA; john@economicrestoration.org

**Abstract:** The current global economic system undermines sustainability efforts such as green infrastructure and circularity, while jeopardizing social and environmental values in rich and poor countries alike. These chronic failures stem in part from a poor understanding of an economy's structure. While many scientists view economic and ecological systems as homologous to one another (i.e., similar structures and processes), an understanding of the foundational components of these systems is lacking. A primary objective of this paper is to refine the understanding of those foundational components, and their interactions. Dozens of ecological processes have functional equivalents in economies, such as succession, evolution, symbiosis, and competition. The central hypothesis here is that three components—diversity, energy, and trade (i.e., resource transfers)—constitute the foundational components of self-regulating economies and ecosystems. A secondary hypothesis is that the interaction of these components regionally, rather than at a global scale, is a limiting factor to the long-term success of sustainability elements such as circularity, green infrastructure, and others. This article evaluates diversity, energy, and trade relative to measures such as net productivity, stability, resource-use efficiency, and biomass (i.e., capital) accumulation. In closing, the concept of economic restoration is summarized, as informed by principles of ecological restoration.

**Keywords:** complexity theory; sustainability; green infrastructure; restoration; resilience; self-reliance; energy neutrality; endogenous economy

## 1. Introduction

At the onset of the 2007 great recession, Simon Johnson of MIT faced a conference of American economists and criticized the failure of existing economic models [1]. In the process, he called upon a powerful analogy: significant progress in health and sanitation came only after society had discovered the structure and inner workings of the human body, and the nature of diseases that threaten it. We lack an equivalent understanding of our economies. As capitalism has grown into a truly global system, monumental efforts have been waged by dozens of nations, and countless individuals and organizations, aiming to resolve our most important social, environmental, and economic challenges. Yet one-quarter of the world's population still suffers moderate to severe food insecurity [2], soil fertility continues declining in essential food-growing regions [3–5], 70% of nations dispose of over half of their waste into landfills [6], 89% of the world's energy consumption is non-renewable [7,8], social justice remains challenged, etc. Considering the lack of sustained social and environmental progress over the past 100 years of global market expansion, we must question if important social and environmental problems can be overcome without first resolving the economic flaws responsible for their creation.

Dozens of ecological phenomena have functional equivalents in economies, such as competition, succession, symbiosis, and evolution, while innumerable components exist in both systems. This paper, though, explores the foundational components common to ecosystems and economies. *Foundational components*, as defined herein, are those capable of producing the greatest change from the slightest shifts, and are most responsible for

maintaining the self-regulating nature of the system. To be considered foundational, a component must function similarly in two or more homologous systems.

The objectives of this article are to define the foundational components of self-regulating (i.e., sustainable) economies, and to examine the basic processes of economic restoration, as inferred by ecological restoration. The terms self-regulating and sustainable are used interchangeably herein, for a self-regulating economy cannot persist with activities that degrade the social and environmental resources it requires for survival. Several hypotheses are provided later, related to the boundary conditions of a sustainable economy, and the interrelationship between diversity, energy, and trade.

### 1.1. Ecosystems and Economies as Homologous Self-Regulating Systems

In studying the structure and functioning of self-regulating systems (e.g., human beings), such as developing a vaccine for polio, it is common to observe treatment responses in homologous systems (e.g., monkeys). Herein, economies and ecosystems are considered homologs; self-regulating systems with very high levels of complexity. A *self-regulating system* is one capable of maintaining key functions and processes in the absence of intervention (i.e., instructions) from beyond the system boundary.

Self-regulating systems exist from cells to ecosystems, where innumerable species and individuals form a web of interconnections to ensure the system's survival in the face of disturbance. For instance, roots in a forest form networks with fungi to facilitate information sharing between trees, so the forest may respond pre-emptively to an external threat such as an insect outbreak [9]. Just as self-regulating ecosystems exist via the free (i.e., unregulated) interactions between consumers, producers, and decomposers, self-regulating economies exist via the unregulated interactions between businesses and consumers across all levels of production and consumption. Systems thinking is one means to examining and understanding these complex homologs.

Systems thinking is a branch of science aimed at understanding the structures and interactions of components within living and non-living systems [10–12]. These structures, and the interactions of components, can be reflected in a system's emergent properties, such as an ecosystem's ability to maintain a *dynamic equilibrium*: a relatively steady state in the presence of outside forces [13,14]. In contrast, many modern economies require constant manipulation of policies and laws, subsidies, bailouts, and other actions to maintain a relatively steady state (i.e., stability), achieve acceptable levels of productivity, and social and environmental assets, and provide other benefits in the face of external forces. That is, these economies are not operating as self-regulating systems, and may in fact be trending toward a state of instability.

Recent examples of instability under the current global economic structure include the 2020 social justice riots in the U.S., followed by the attack on the U.S. capital; the economic collapse in Sri Lanka in 2019–2022; Argentina's economic crisis of 2023; the global economic turmoil following the Russia–Ukraine conflict of the 2022–present day; the Israel–Hamas conflict of 2023; the mass exodus of migrants from developing to developed countries; eleven unemployment spikes (i.e., >6%) in the U.S. since 1950; and the unprecedented rise in cost of living globally. Meanwhile, great degrees of wealth have become concentrated into fewer hands [15], a condition some believe threatens rather than ensures a steady state [16,17].

### 1.2. The Structure of Sustainable (Self-Regulating) Economies

Classic definitions of sustainability focus on efforts that improve social, environmental, and economic (i.e., industrial) conditions. This can include the recent circularity efforts, with goals such as zero waste and 100% renewable energy. However, an understanding of an economic structure that would naturally support sustainability, circularity, and related goals is lacking.

The term *highly functional* is used herein to describe an economy that can sustain the production and distribution of goods and services, without jeopardizing the social and

environmental assets upon which it relies. This includes desired economic benefits such as relatively stable employment, profitability, adequate real wages, and the ability to meet basic consumer needs in the face of disturbances. That is, a highly functional economy is relatively stable, a function of resistance and resilience [18,19]. Understanding the structure of a highly functional economic system, though, requires us to look beyond unimportant details [20], such as the system's context. That context includes capital management systems such as capitalism, often blamed for poorly functioning economies. However, great resource inefficiencies, and degradation of social and environmental assets, have arisen in capitalist, socialist, and communist systems alike, suggesting that a country's capital management system is not a prerequisite to a highly functional economy. A peculiar phenomenon, though, has arisen over the evolution of global market capitalism. With each novel system, the duration of each new system has shortened (Table 1). Throughout this evolution, abundant social and environmental challenges have persisted, while the actual structure of the economic system has remained obscured.

**Table 1.** Epochs of capitalism. Data from Arrighi [21] and Braudel [22], with duration indicating years between signal crises. Signal crises occur in periods of comparatively stable governance.

| Epoch of Capitalism | Dominant State | Approximate Time Period (AD) | Duration (Years) |
|---|---|---|---|
| Mercantile | City-states of Northern Italy (Genoa) | 1340–1560 | 220 |
| Mercantile | Dutch | 1560–1740 | 180 |
| Classical | Great Britain | 1740–1870 | 130 |
| Classical | United States | 1870–1930 | 60 |
| Keynesian | United States | 1940–1970 | 30 |
| Global Market | ? | 1970–? | ? |

1.2.1. Chronic Social and Environmental Challenges of the Current Economic Structure

The current epoch of global market capitalism (GMC) allows for levels of resource consumption exceeding any previous epoch, corresponding with a litany of social and environmental degradation [16,23–25]. Degradation occurs in direct forms such as social and environmental impacts, and indirect forms such as supply chain inefficiencies, where small shifts in demand can produce an increasing magnitude of interruptions for manufacturing and extractive industries at the opposite end [20].

As supply chains lengthen, or as the number of entities (i.e., links) in the chain increase, inefficiencies arise with rising uncertainty and risk for producers and consumers. To buffer against risks, producers and distributors accumulate *safety stock*, generating and storing more products than what consumers demand. Safety stock translates not only to increased financial costs of production, but increased energy and material waste. For instance, when short-lived goods (e.g., bananas and fast fashion) become safety stock, much of the stock becomes unsalable before reaching customers, leading to physical, energy, and financial waste. Another example of supply chain inefficiency is U.S. consumers who purchase food originating from an average of 1500 miles away [26], sixteen million tons of which are wasted annually [27]. Hawaii is a notable example, which ships cattle 2000 miles to the mainland for processing. The preferred shipping method is cargo planes [28], one of the least energy-efficient means of transportation [29].

Uncertainty aside, consumers at the end of long supply chains often underestimate the unintended social and environmental impacts of production and extraction at the other end (i.e., externalities). Classic examples include water pollution caused by paper manufacturing, or fertilizer run-off from industrial-scale agriculture. In 2023, a train derailment in Ohio (U.S.A.) caused an unprecedented vinyl chloride spill, an externality related to PVC pipe consumption [30]. Species extinction is another chronic externality, the leading cause being economic activity [31].

Corporate or national systems intended to improve social, economic, and environmental conditions, yet lack persistence, and thus are intrinsically unsustainable. A few

examples are provided here. Victoria's Secret's corporate sustainability commitment states "We strongly believe that the quality of our merchandise begins with the treatment of the people who create our products [32]". However, during Sri Lanka's recent economic collapse, Victoria's Secret asked Sri Lanka's largest textile supplier, Mass Holdings, to shift production to their Bangladesh facility (a lower labor market), or cut costs in their Sri Lanka facilities by 30% [33]. In Germany, decades of work to reduce their carbon footprint was threatened when the country took steps to bring coal-powered plants back on-line [34]. In the U.S., decades of environmental protections were undone by a 2020 presidential signature, then partially restored by a 2022 presidential signature, and weakened again by the supreme court in 2023. Despite significant agriculture advancements in the 20th century, 33 million acres of the U.S. Corn Belt have zero topsoil remaining [35]. These persistent externalities, I posit, stem not from the form of capital management system (e.g., capitalism, socialism, or communism), but from the structure of one's economic model.

### 1.2.2. Economic Structure as a Function of Foundational Components and Context

Past research has indicated that the structure of an economy or ecosystem is defined by only energy flows [36–38]. While energy is certainly essential to the structure and functioning of economies and ecosystems, the central hypothesis here is that three foundational components—diversity, energy, and trade (i.e., resource transfers)—comprise the core structure of economies and ecosystems. These components form predictable interactions with one another, while being connected via feedback loops with an economy's social-environmental-industrial (SEI) context. An economy's *social context* includes its cultural values, models of governance, desired levels of productivity, etc. Its *environmental context* includes soil fertility, precipitation, the quality of natural resources, etc. Whilst, its *industrial context* includes technology, the quantity and connectivity of trade centers, infrastructure, etc. The full structure of an economy is defined herein as the balance of its foundational components—diversity, energy, and trade—in relation to the SEI context in which they exist (Figure 1).

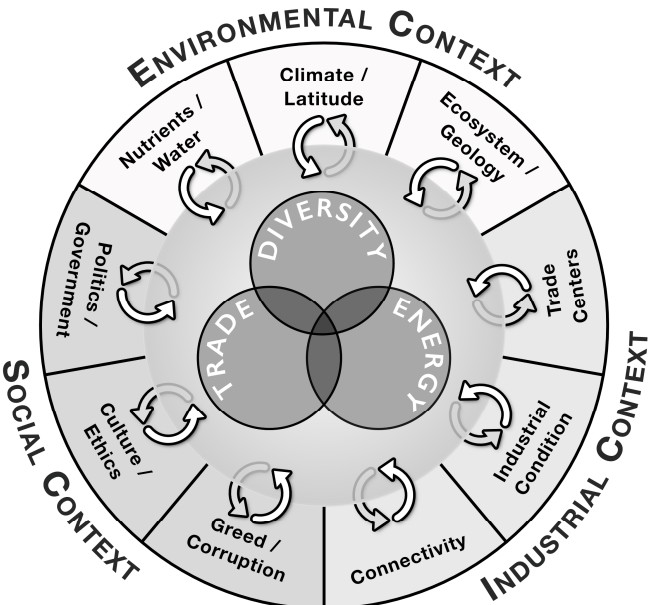

**Figure 1.** The structure of an economy consists of its foundational components—diversity, energy, and trade—and its context. A few contextual factors are pictured in the outer ring, with chasing arrows indicating feedback loops with the system's foundational components.

One's SEI context may be manipulated to the benefit of humans, their economies, and the environments in which they exist. The Llanos region of Colombia is one example, where human ingenuity (social context) has increased economic productivity and diversity [39].

Another example is Salinas de Guaranda, Ecuador, whose social and industrial context has been managed since the 1970s to increase the productive capacity and diversity of a high-elevation Andean economy [40,41].

*1.3. Relationships between Productivity Measures and Sustainability*

Relationships between economic productivity measures and sustainability are indirect but meaningful. Gross domestic product (GDP), for instance, does not account for the degradation of social and environmental conditions that result from production and extraction processes. Two productivity measures that do provide a more realistic account of system degradation, due to their positive correlation with resource-use efficiency, are productive capacity and *net* domestic production. A nation's *productive capacity* reflects its ability to produce goods and services in the future, and that capacity increases when inputs are used more efficiently (e.g., a bushel of wheat produced with less nitrogen). Such efficiencies are gained via the skill of labor, technology, political and economic stability, and other factors.

A nation's *net domestic production* (NDP) also rises with energy- and materials-use efficiency. Akin to net primary production in ecosystems, an economy's NDP accounts for system maintenance, such as the depreciation of factories, roads, and other assets. However, traditional NDP measures often fail to account for the degradation of natural resources that accompany production processes, hence NDP overestimates productivity. For example, restoring the 33 million acres of degraded farmland topsoil in the U.S. Corn Belt is estimated at U.S. $1.2 trillion [42]. That expense was unaccounted for in the years that farm outputs contributed to GDP. To more fully account for system maintenance, a measure of *real NDP* is provided here: economic productivity after accounting for the depreciation of capital assets, social assets, and ecological assets. NDP used herein represents this definition.

Indirect measures of NDP, such as valuing ecosystem services (e.g., clean water from a forested watershed), attempt to reflect the environmental costs of production. A challenge with many ecosystem service models is that they do not reflect natural supply–demand dynamics. Instead, they rely on hypothetical markets to quantify the value of services such as clean water, or the value of a species, etc. [43,44].

Real NDP must also account for the quantity of energy consumed in support of an economy's per capita productivity (Table 2). This measure reflects an economy's ability to maintain per capita productivity in a future of limited and/or uncertain energy supplies. From this measure, Iceland ranks 2nd, Italy ranks 40th, and the U.S.A. ranks 65th. Confounding variables certainly affect a nation's per capita GDP on a unit energy basis. For instance, Luxemburg has a sizable private banking industry, and Iceland harbors large supplies of geothermal energy, relative to their populations.

**Table 2.** Per capita GDP on a unit energy basis. A partial list of countries is presented. Energy consumption data are available for 72 countries.

| Rank | Country | Per Capita GDP on Unit Energy Basis * | GDP per Capita (USD) | GDP 2021 (Billion USD) | Population 2021 | Avg Energy Consumption ** |
|------|---------|---------------------------------------|----------------------|------------------------|-----------------|---------------------------|
| 1 | Luxembourg | 37,108 | $133,590.15 | $85.51 | 640,064 | 3.6 |
| 2 | Iceland | 13,746 | $68,727.64 | $25.60 | 372,520 | 5.0 |
| 3 | Cyprus | 8782 | $22,832.46 | $28.41 | 1,244,188 | 2.6 |
| 4 | Ireland | 6678 | $100,172.16 | $504.18 | 5,033,165 | 15.0 |
| 5 | Latvia | 5874 | $21,148.16 | $39.85 | 1,884,490 | 3.6 |
| 6 | Estonia | 4902 | $27,943.70 | $37.19 | 1,330,932 | 5.7 |
| 7 | Slovenia | 4577 | $29,291.40 | $61.75 | 2,108,079 | 6.4 |
| 8 | Denmark | 4250 | $68,007.71 | $398.30 | 5,856,733 | 16.0 |
| 9 | Lithuania | 3954 | $23,723.34 | $66.45 | 2,800,839 | 6.0 |
| 10 | Switzerland | 3593 | $91,991.58 | $800.64 | 8,703,405 | 25.6 |

**Table 2.** *Cont.*

| Rank | Country | Per Capita GDP on Unit Energy Basis * | GDP per Capita (USD) | GDP 2021 (Billion USD) | Population 2021 | Avg Energy Consumption ** |
|---|---|---|---|---|---|---|
| 12 | New Zealand | 2439 | $48,781.09 | $249.89 | 5,122,600 | 20.0 |
| 13 | Norway | 2286 | $89,154.30 | $482.18 | 5,408,320 | 39.0 |
| 16 | Finland | 1937 | $53,654.77 | $297.30 | 5,541,017 | 27.7 |
| 19 | Sweden | 1212 | $61,028.76 | $635.66 | 10,415,811 | 50.3 |
| 20 | Portugal | 1135 | $24,567.50 | $253.66 | 10,325,147 | 21.6 |
| 22 | Singapore | 881 | $72,794.02 | $396.99 | 5,453,566 | 82.6 |
| 23 | Belgium | 844 | $51,247.00 | $594.10 | 11,592,952 | 60.7 |
| 24 | Greece | 804 | $20,192.61 | $214.87 | 10,641,221 | 25.1 |
| 26 | Netherlands | 754 | $57,768.06 | $1012.85 | 17,533,044 | 76.6 |
| 30 | United Arab Emirates | 509 | $44,315.60 | $415.02 | 9,365,145 | 87.1 |
| 31 | Australia | 464 | $60,443.21 | $1552.67 | 25,688,079 | 130.3 |
| 32 | Sri Lanka | 441 | $4013.69 | $88.93 | 22,156,000 | 9.1 |
| 36 | Ecuador | 328 | $5965.14 | $106.17 | 17,797,737 | 18.2 |
| 37 | United Kingdom | 284 | $46,510.32 | $3131.38 | 67,326,569 | 163.6 |
| 39 | Spain | 243 | $30,103.50 | $1427.38 | 47,415,750 | 123.8 |
| 40 | Italy | 239 | $35,657.45 | $2107.70 | 59,109,668 | 149.4 |
| 42 | France | 190 | $43,658.98 | $2957.88 | 67,749,632 | 229.8 |
| 43 | Germany | 174 | $51,203.50 | $4259.93 | 83,196,078 | 293.8 |
| 46 | Canada | 167 | $51,988.03 | $1988.34 | 38,246,108 | 310.8 |
| 49 | Argentina | 134 | $10,636.11 | $487.23 | 45,808,747 | 79.2 |
| 50 | Malaysia | 119 | $11,109.26 | $372.98 | 33,573,874 | 93.7 |
| 51 | South Korea | 117 | $34,997.86 | $1810.96 | 51,744,876 | 300.3 |
| 53 | Japan | 95 | $39,312.68 | $4940.88 | 125,681,593 | 412.0 |
| 55 | Philippines | 74 | $3460.53 | $394.09 | 113,880,328 | 46.9 |
| 57 | Bangladesh | 62 | $2457.93 | $416.27 | 169,356,251 | 39.5 |
| 58 | Mexico | 61 | $10,045.69 | $1272.84 | 126,705,138 | 164.7 |
| 62 | Ukraine | 54 | $4568.92 | $200.09 | 43,792,855 | 84.9 |
| 64 | Vietnam | 36 | $3756.49 | $366.14 | 97,468,029 | 103.2 |
| 65 | United States | 32 | $70,248.69 | $23,315.10 | 331,893,745 | 2172.0 |
| 66 | Brazil | 25 | $7507.15 | $1608.98 | 214,326,223 | 304.2 |
| 67 | Indonesia | 20 | $4332.70 | $1186.09 | 273,753,191 | 212.0 |
| 68 | Russia | 17 | $12,400.06 | $1778.78 | 143,449,286 | 747.5 |
| 71 | China | 3 | $12,556.36 | $17,734.10 | 1,412,360,000 | 3708.6 |
| 72 | India | 3 | $2256.59 | $3176.30 | 1,407,563,842 | 886.6 |

* Data from World Bank, Enerdata, and BP. Energy in Mtoe. GDP in USD (production approach); ** Average of Enerdata and BP data, in Mtoe.

## 2. Materials and Methods

Evaluating concepts such as the foundational components of highly complex economies, or the basic operational unit of an economy (i.e., boundary conditions) requires methods other than traditional hypothesis testing. The methods herein are drawn from systems thinking, in which theoretical concepts are used to explain and predict natural phenomena [45]. This includes general systems theory, which explores concepts such as system boundaries and self-organization. For these reasons, an interpretive review was used, including the extraction and synthesis of data from selected studies.

Data were evaluated using a crosscutting approach, where the reviewer evaluates systemic patterns between two homologous systems [46]. Similar methods are used in homologs such as monkeys and humans. For instance, Rhesus Macaque (*Macaca mulatta*) was the homolog used to develop the polio vaccine for *Homo sapiens* [47]. Herein, economies and ecosystems are presumed homologous, across which several classes of information were compared: system subcomponents, theories, principles, processes, material and energy flows, concepts, functions, and other factors. Data from aquatic systems were excluded, as they differ fundamentally from terrestrial systems via energy flow and trophic pathways [48]. To understand the relationship between energy and biological diversity, this paper focused on a form of plant-available energy reported widely in the literature,

which can be transferred across trophic levels: nitrogen, an energy metabolite from which glucose and other energy compounds are formed.

The interpretive review of ecological and economic literature included semantic scholar, PrimoSearch (i.e., a network of over 400 databases across economic and ecological disciplines), Econ Papers (https://econpapers.repec.org, accessed on 28 June 2023), and others. Relevant texts from prominent economists (e.g., David Ricardo, Adam Smith, Karl Marx, John Maynard Keynes, Alan Greenspan, Joseph Schumpeter, Milton Friedman, Joseph Stiglitz, Thomas Piketty, and Herman Daly), and from the career works of ecologists and evolutionary biologists, were also evaluated.

## 3. Results and Discussion

Abundant literature exists on the interactions between energy and ecological diversity. However, little research exists on the interactions between energy and economic diversity, or trade and economic diversity. From this integrative review, three foundational components appear to exist in both ecosystems and economies, forming equivalent interactions regardless of the context in which they occur: diversity, energy, and trade. These components, and their interactions, are summarized below, with more detail in Giordanengo [42]. Due to the natural interactions between diversity, energy, and trade, each section below will integrate some relevant information from other foundational components as appropriate.

### 3.1. Diversity, and Its Influence on Ecological and Economic Systems

*Biological diversity* has many aspects, including the quantity and distribution of species within a habitat, and the variety of habitats in an ecosystem. *Economic diversity* defined herein is simply the quantity and distribution of businesses within an economy. Economic diversity is evaluated here in terms of the economic and environmental benefits it imparts, drawing upon equivalent benefits in ecosystems. For example, ecosystems with greater plant diversity are capable of harnessing more energy from a given environment, and generating higher net productivity, compared to less diverse systems [19,36,37,49–54]. Diverse plant communities not only support a high diversity of insects, for instance, but a greater biomass of insects [55–57], from which a greater abundance of insectivores may be sustained, etc.

Research on the relationship between economic diversity and energy is sparse. Paul Templet compared national economic diversity data with the annual energy throughput of twelve countries, concluding that a positive correlation exists between economic diversity and energy-use efficiency [58]. One example of the diversity–efficiency relationship is the excess waste heat and other byproducts generated by lumber mills, which may be used to warm nearby buildings or generate biomethanol. In one Alaska (U.S.A.) community, the energy gained from a nearby mill's waste heat saved an estimated U.S. $91,500 per year [59]. At a mill in Bellemont, Arizona (U.S.A.), selective logging and milling operations produce five residual products for regional consumers, including two forms of biomass fuel [60], in part supplying Novo Power, a 27 MW biomass powerplant in northern Arizona. These specific efficiencies do not occur in an economy without a mill.

Waste heat is produced by all manufacturing and resource extraction industries. Hence, economies lacking those industries (i.e., consumption needs are met with goods extracted or manufactured outside the economy) forgo a significant form of waste heat to offset aggregate energy consumption. In the U.S., over 1500 trillion BTUs of waste heat are available for recovery from manufacturing and mining industries annually [61]. As heating the average U.S. home consumes 75,000 BTUs of energy annually, about 20 billion homes could be supplied by industrial waste heat—more homes than exist on earth. However, under the current global economic structure, industrial activities associated with U.S. consumption are concentrated in relatively few and homogenized economic spaces across the globe (i.e., diversity and trade components), often requiring exogenous energy inputs (energy component) into those spaces. Additional exogenous energy is consumed to ship

manufactured goods (trade and energy components) into economies with low diversity in their manufacturing and resource extraction sectors.

In ecosystems and economies, more diverse systems are not only more energy efficient, but also more stable than low-diversity systems [62–70]. While energy efficiency and stability are directly related to high NDP and productive capacity, the relationship between diversity and NDP is less direct. Diversity in an economy promotes innovation, via knowledge and technology spillovers across sectors [71,72]. Innovation has long been recognized as a key mechanism in productivity gains, which stimulates economic growth [73].

Some economic research does show a positive correlation between diversity and economic productivity [58,74], though further research is needed. In ecosystems, a positive correlation between diversity and productivity is well known, summarized by the intermediate disturbance hypothesis [75], and to be discussed later. In economics, confounding variables in the diversity–productivity relationship arise when studies account for gross productivity, not NDP, or when they are limited in their temporal and spatial analysis, fail to use per capita GDP on a unit energy basis, or compare results across varying SEI contexts, each with a unique productive potential. A more fundamental concern is a lack of accounting for the multiple forms of diversity that exist in economies and ecosystems. Eight forms of economic diversity are described in Giordanengo [42], of which seven with functional equivalents in ecosystems. Five are described here.

### 3.1.1. Richness and Evenness

One of the most basic measures of biodiversity is the number of species present in an area, what ecologists refer to as *species richness*. When richness is evaluated in conjunction with the relative "dominance" of each species in the system (i.e., *evenness*), a measure of *species diversity* is provided. For instance, a meadow with a hundred species of plants is more *diverse* than a meadow with ten, assuming each species contributes equally to the biomass (i.e., capital) of the system. Species diversity is quantified by Shannon's Diversity Index [76] and other indices. Similar indices are used to measure economic diversity, only with businesses taking the place of species.

### 3.1.2. Functional Diversity

Every species is part of a functional group, such as producers (e.g., plants), consumers (e.g., herbivores), or decomposers (e.g., fungi), referred to as *trophic levels* in ecology. *Functional Diversity* is simply a diversity of trophic levels, and is a driver of other forms of diversity [77,78]. Systems with high functional diversity are also capable of producing more total biomass [79], as they contain a greater diversity of food chains.

In an economy, functional diversity increases when it harbors, for instance, a network of farms, mills, retailers, steel manufacturers, high tech, and recycled material processors. Economists simply use different nomenclature than ecologists, such as sectors, industry groups, and industries, in production-based hierarchies like the Nomenclature des Activités Économiques dans la Communauté Européenne (NACE), or the North American Industry Classification System (NAICS). A single-function economy, such as one in which only apples are sold, has very low functional diversity. On the contrary, an economy with a high degree of functional diversity (e.g., many sectors exist) has the potential to increase the diversity of various industry groups, each harboring multiple industries, of which each houses at least one business. In highly complex self-regulating systems, it is uncommon for a given functional group to be represented by just one business, or just one species.

### 3.1.3. Functional Redundancy

Within a given functional group, such as insectivorous birds, multiple species often exist (e.g., nuthatches, chickadees, and juncos), each contributing to *functional redundancy* within the group. Should the population of one species decline, other species can fill the void. In a review of 423 research articles, Biggs et al. [80] found that higher levels of functional redundancy were correlated with higher levels of ecosystem resilience and stability.

Economies are similarly diversified when multiple businesses exist to, for instance, manufacture or install wind turbines, or when multiple industries exist to generate electricity (e.g., biomethanol, hydro-electric, or solar). Should one industry falter (e.g., solar), other energy systems remain to meet demand. The American auto industry is an example of low functional redundancy. The industry became dominated by just three companies in the 1980s, then collapsed in response to a variety of external forces [81].

### 3.1.4. Regional Diversity

Biological diversity also exists at a landscape scale [82]—*regional diversity*—when multiple habitat patches exist in an ecological space, each shaped by differences in soils, moisture gradients, and other factors [83,84]. Patches are further diversified by disturbances that allow a variety of successional communities to exist there. Each successional community exhibits a unique species assemblage, reflecting the amount of time since the disturbance occurred (e.g., a fire occurred there 10 years ago) and the severity of the disturbance that occurred (e.g., a low-severity fire). This disturbance–diversity process is documented well by Beever [85], and regional patch diversity is a multiplier of other forms of diversity [86]. Regional diversity also exists in economies that harbor a patchwork of cities, farm communities, centers of wood production or metal recycling, energy production, and various retail centers. Based on the nature of regional diversity in ecosystems, we might expect regional diversity in an economy to be a multiplier of other forms of economic diversity.

Despite the benefits, diversity cannot simply be added to an economy, without considering its interactions with other foundational components. Evidence from economics and ecology indicates, however, that diversity may be stimulated (or suppressed) by the management of trade and energy.

### 3.2. Relationships between Energy and Diversity

Energy is referred to here in terms of the forms cycled within an ecosystem, such as carbohydrates (i.e., biochemical energy), or an economy, such as solar power, biogas, and coal. Economists refer to these "usable" energy forms as *primary energy*, used in energy balance equations [87]. How energy flows through economies and ecosystems affects system diversity. Diversity, in return, has an impact on energy-use efficiency.

In ecosystems, energy is often studied with respect to its flows through *trophic levels* (i.e., producer and consumer levels in a biomass pyramid). At every trophic level, energy is lost in support of growth and metabolism [37,88]. In the ecological literature [36,53], and limited economic literature [58,89], more diverse systems are reported to capture a greater quantity of energy than less diverse systems. In other words, with a fixed quantity of energy, more diverse systems support a greater level of net productivity.

While research on energy flows in economies is found in the work of Alfred Lotka [90,91], Bruce Hannon [37], Amory Lovins [92], and others, studies on the interaction between energy flows (i.e., exogenous vs. endogenous energy) and economic diversity are absent. In ecosystems, the moment primary producers transform sunlight into biochemical energy, energy becomes endogenous to the system. *Endogenous energy* is defined herein as energy consumed in the same ecological or economic space in which it is produced. To be endogenous, energy must also be produced by extant entities.

Consumers in terrestrial ecosystems obtain nearly 100% of their biochemical energy from endogenous sources. Species in each trophic level (e.g., lion) obtain energy only from the level below (e.g., gazelle), which obtains energy only from primary producers (e.g., plants). The system's biochemical energy is limited by the quantity of incoming solar radiation, which varies across the globe. That is, photosynthetically active radiation (PAR) is part of an ecosystem's environmental context, and does not become endogenous until converted to biochemical energy.

In economies, endogenous energy includes solar-generated electricity, hydropower, biofuels, etc. In contrast to ecosystems, over 80% of the energy in national economies is exogenous, mostly in the form of fossil fuels. *Exogenous energy* is that which is produced

in a different place than where it is consumed, or by entities that no longer exist. Fossil fuels are one example, created hundreds of millions of years ago by extinct carboniferous forests, and usually imported into an economy from thousands of miles away. Electricity generated by wind turbines in Wyoming (U.S.A.), and then transported to New York (U.S.A.), is also exogenous to New York's economy. Imported agricultural products (e.g., cattle, grains, and fruits) also represent exogenous energy, in the form of carbohydrates and other embodied energy.

In some cases, exogenous energy can diminish an economy's capacity to produce endogenous energy. Sri Lanka, with very few fossil fuel reserves, has operated as a net energy importer for over 50 years. Today, about 25% of its GDP is expended on imported fossil fuels [93], contributing to high trade imbalance and other financial imbalances [94]. In an interview with Shihan Diddeniya in 2023, I was informed that Sri Lanka has enough renewable energy potential in wind, solar, hydroelectric, and biomass to supply over 100% of its total energy needs [95]. However, following its 2019 economic crash, Sri Lanka's capacity to develop these endogenous energy resources has diminished [96].

The terms exogenous and endogenous apply also to material resources such as protein, steel, and water. For instance, nitrogen is cycled between the atmosphere and the biosphere continuously, but does not become endogenous to an ecosystem until fungi and bacteria convert $N_2$ into plant-available forms such as ammonium, an energy metabolite [97]. Once assimilated by the biotic system, nitrogen, biochemical energy, and other resources are then available to be cycled across trophic levels within a food web.

### 3.2.1. Interactions between Exogenous Energy and System Diversity

Regions of earth receiving higher PAR tend to have higher biological diversity, as PAR peaks at the equator and diminishes toward the poles. This includes research by Wright [98] on 24 islands across the globe, noting that larger islands near the equator are richer in plants and birds than those toward the poles. At the surface, this appears to infer a positive correlation between energy and diversity. However, at least two confounding variables must be considered. The diversity gradient from the equator to the poles is more closely related to precipitation patterns than PAR [99]. Second, equatorial ecosystems were minimally impacted by the most recent glaciation events, compared to ecosystems nearer to the poles. The Laurentide ice sheet, for example, eliminated a wide range of plant communities from vast tracts of North America for thousands of years.

PAR is simply part of an economic or ecological context that becomes endogenous energy that can then be transferred across trophic levels within a specific context. Of interest here is the influence of cross-boundary transfers (e.g., between two economies, or between two ecosystems) of primary or biochemical energy on system diversity, regardless of the system's context.

Results from over 190 studies across the globe indicate that transfers of plant-available nitrogen (i.e., an energy metabolite) into an ecosystem are correlated with decreased plant community diversity [100,101]. Though economic studies on the relationship between exogenous energy and diversity are lacking, some trends are evident. Over the past 120 years, an exponential rise in exogenous energy consumption has occurred in the U.S. (Figure 2), a trend seen in many industrialized economies [29]. Diversity data from U.S. agriculture and manufacturing over this same period are presented below.

*Agriculture*: The 120-year rise in exogenous energy coincides with the extinction of over five million farm businesses [102]. One example comes from Oakdale in northeast Nebraska (U.S.A.), dominated today by corn (Figure 3). In the 1870s, Oakdale's economy exhibited high regional diversity, richness, and functional redundancy: dozens of different crops, barber shops, mercantilists, doctors' offices, manufacturers, etc. [103]. Trees—not children—were growing up through the playground equipment when I visited Oakdale in 2011. In 2020, "no entry" signs were posted at the entrances of schools and churches, and 20% of the population lived in poverty [104].

Besides the social and economic degradation in Oakdale, its homogenous agricultural landscape is more prone to external forces, such as those that produced the Potato Famine [105,106], the U.S. Dust Bowl, the 1970s southern corn leaf blight, etc. Additionally, future climate scenarios predict that the corn belt will experience additional stressors including increased heat stress, higher peak floods, and longer growing seasons [107]. Concurrently, US farmers are expected to experience rising costs of exogenous inputs and declining profitability [108], impairing their ability to address future stressors.

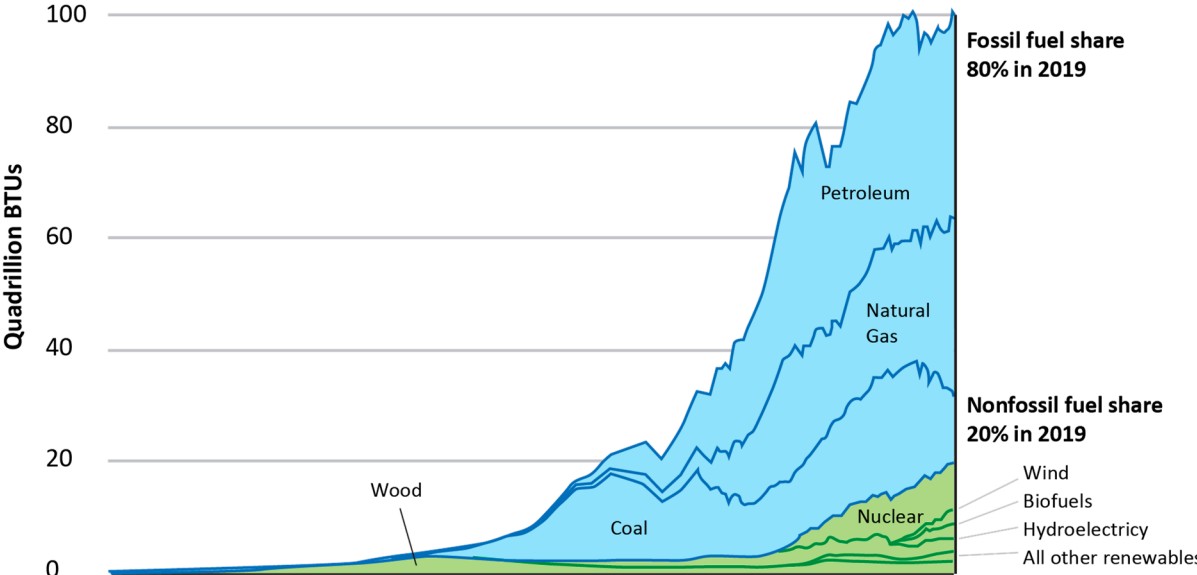

**Figure 2.** Energy consumption in the United States of America, from 1776 to 2020. Source: EIA, public domain [109].

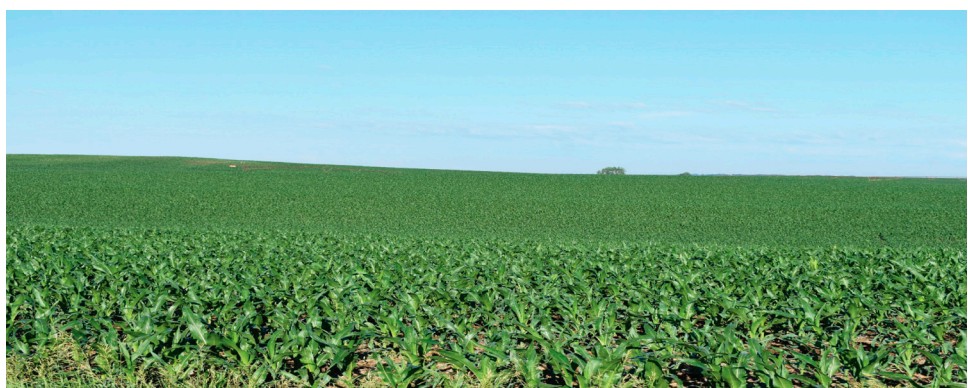

**Figure 3.** Monoculture of corn near Oakdale, Nebraska, U.S.A.

*Manufacturing*: Over one hundred automobile manufacturers existed in the U.S. in the 1920s [110]. By the early 1950s, the consolidation of auto manufacturing around Detroit, MI, made this metropolis one of America's wealthiest cities [111], its prosperity hinging on the success of a single industry. In that same decade, competition began rising from Europe and Asia, while America's involvement in the Korean War redirected steel supplies and labor away from Detroit [112]. By 1980, the U.S. auto industry was dominated by three companies [83], with low richness. By the late 1990s, Detroit was described as the U.S. murder capital, with half of its real estate abandoned.

Other extinctions occurred in U.S. manufacturing in the past 120 years, with the decline in business richness estimated at 70%, on a per capita basis [42]. This amounts to a loss

of about 700,000 businesses, including an estimated 318,000 bakeries, over 500 wood and paper mills, 100s of textile mills, and 100s of grain mills.

Though the inaccuracy and incompleteness of historical economics data make statistically robust analyses difficult, a correlation between diversity and exogenous energy does appear likely. For instance, exogenous energy transfers into the U.S. economy remained low between 1849 and 1899, relative to the total energy consumed, and manufacturing diversity climbed from 530 to 672 businesses per 1000 citizens [113]. This increase occurred despite the decade-long global economic recession of the 1890s. As exogenous energy consumption increased between 1900 and 1929, manufacturing diversity declined from 291 to 191 businesses per 1000 citizens. The difference in the magnitude of diversity gains and losses reflects a shift in the definition of "manufacturer" at the turn of the 20th century. The exogenous:endogenous energy ratio has continued rising since the 1930s Great Depression, with commensurate declines in the diversity of manufacturing and agricultural industries.

### 3.2.2. Challenges in Quantifying Economic Diversity

The lack of historical business diversity data [114] is one challenge to quantifying diversity over time. For instance, non-commercial home bakeries and artisan workshops of the 1800s were unrecorded in U.S. census data. Further, business data collected in the 1800s were biased toward specific industries, or toward urban areas where significant economic activity occurred, therefore underreporting diversity at regional scales. Other complications arise via variations in industrial classification systems over time. NACE did not exist until 1970, and NAICS was not adopted until 1997 [115]. As these classification systems evolved, so did definitions of "industry" and other basic measures.

Diversity measures like Shannon's Diversity Index provide another limitation, as they account for only two aspects of diversity, and suffer some statistical limitations [18,69]. The more recent Economic Complexity Index suffers similar limitations, and accounts for only the diversity of export-based industries [116]. Economic diversity data also ignore regional diversity patterns, and often aggregate data at the county and industry level. As a result, such studies report lower diversity in rural agricultural areas than in metropolitan areas within the same region, and can under-represent functional redundancy and regional diversity. Other limitations include the relatively short periods of economic diversity studies (e.g., 1–3 decades), or they compare regional diversity to a national benchmark, regardless if that benchmark has declined. Complications also arise from the fact that new industries have arisen over time.

### 3.2.3. Quantifying Net Changes in Economic Diversity

Considering the above limitations, we might ask if *net* economic diversity (i.e., the difference between gains and losses) actually increased in the U.S. over time, not decreased. For instance, thousands of new businesses have emerged in high technology, services, and others over the past century, offsetting diversity losses in manufacturing and agriculture. At the same time, new entrants in high tech (e.g., Google, Amazon, Microsoft) have followed similar diversity patterns as in manufacturing and agricultural: a tendency toward homogenization. The retail industry exhibits similar patterns, as "Big box" stores (e.g., Target, Walmart, Home Depot) have displaced thousands of businesses in some countries, and franchises such as McDonalds (37,000 storefronts globally) produce direct and indirect impacts on diversity. When standards of operation are controlled by a single entity, the tendency is for franchisees—or chains—to source tomatoes, buns, and beef from large homogenized manufacturers, placing downward pressure on diversity in agriculture and food processing. In the U.S., 773,000 franchises exist [117], half of which are in food service.

To gain a fuller understanding of diversity, and to better quantify diversity changes over time, more comprehensive measures are needed. Those measures must integrate not only richness and evenness, but functional redundancy, matrix diversity, and other diversity elements into a value that reflects system-wide diversity.

### 3.2.4. Relationships between Technology and Economic Diversity

Several other variables were evaluated as possible drivers of economic diversity in Giordanengo [42], including technology, one element of an economy's SEI context. For example, abundant technological advances occurred during the Industrial Revolution, such as steam power, often attributed to the homogenization of manufacturing and agriculture [118]. We might conclude, then, that technology (not energy) was the cause of economic diversity declines in that period. However, steam power was invented six centuries earlier in Greece [119]. Not until great quantities of dense energy were discovered (e.g., coal in England) did the productive potential of steam engines become realized [29,118]. It was the coal-powered steam engine that allowed the industry to manufacture larger and more precise equipment, including high-pressure boilers, a limiting factor to larger steam-powered generators [22,120]. Large-scale generators were essential to the mass production of larger machinery such as tractors and trains, which further stimulated homogenization in affected industries.

Considering the interactions between energy and diversity in both economies and ecosystems, this hypothesis is proposed: exogenous energy transfers into self-regulating ecosystems and economies, causing a reduction in diversity in those systems.

### 3.3. Trade (i.e., Resource Transfers) in Economies and Ecosystems

In ecosystems, resources such as fat, micronutrients, and protein begin their transfer to higher trophic levels when plants convert sunlight, $CO_2$, and water into biochemical energy. A notable difference between resource transfers in modern economies and in ecosystems is the origin of resources being cycled through each system. Nearly 100% of the biomass consumed and cycled within an ecosystem is produced within the system—it is endogenous, a product of primary production. In other words, the further one travels from a given habitat patch (e.g., Mediterranean scrub community in Liguria, IT), the less likely they are to encounter energy and other resources from that community. That is, the degree of resource transfers between trophic levels within an ecosystem is far greater than the resource transfers across ecosystem boundaries.

The opposite condition occurs in most modern economies. As one travels from national to local centers of consumption—nation, region, town, or home—they are more likely to encounter resources from other towns, regions, and nations. From U.S. Census Bureau import–export data, about 32% of goods consumed in the U.S. are imported, based on dollar value [42]. However, because most U.S. exports are of high value (e.g., spacecraft, airplanes), relative to imports (e.g., dolls, cookware), the quantity of imported goods is higher than 32%. In most U.S. regions, the percentage of imported goods is estimated at 60%, while many urban areas import closer to 99% of physical goods [42]. Similar trends exist globally, as the portion of global GDP attributed to foreign trade has increased from 36% in 1979 to 60% in 2019 [120]. This coincides with an approximate 100% increase in energy consumption [29].

Historically, economists have partitioned trade into three segments: domestic, foreign, and carrying [121,122]. The domestic economy (i.e., *home trade*) is the transfer of goods and services within a nation, which was the dominant form of trade through the 18th century [123]. *Foreign trade* is the transfer of goods and services across national borders. A limiting factor to foreign trade before the 18th century was fossil fuels, which were needed to increase the efficiency of long-distance transportation [29]. The *carrying trade* (i.e., carry trade) is the most nebulous echelon of global market capitalism, and includes multiple financial mechanisms used to facilitate foreign trade.

Adam Smith posited that the strength of a nation's domestic trade—not its foreign or carrying trade—forms the basis of its wealth [122]. That foundation, he argued, is sustained by a country's manufacturing and resource extraction sectors. A conceptual representation of Smith's trade paradigm, similar to the biomass pyramid of ecosystems, is the balanced trade pyramid (Figure 4). The majority of trade is domestic, and over

fifty percent of domestic production is generated by agriculture, manufacturing, and other "productive" industries.

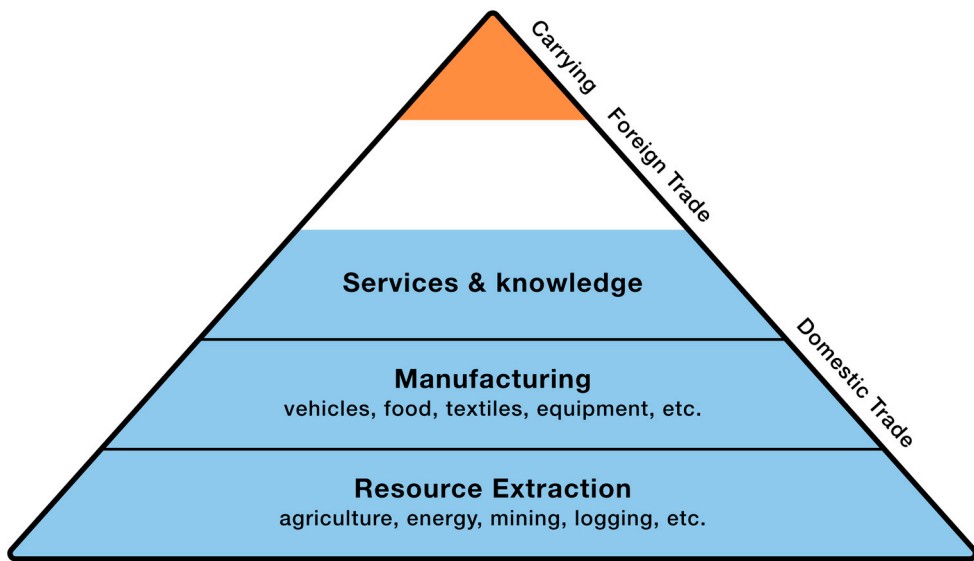

**Figure 4.** Balanced trade pyramid. The relative contribution of the domestic, foreign, and carrying trades in an economy. Relative GDP of various sectors within the domestic economy.

Today, U.S. agricultural and other extractive industries comprise less than 4% of GDP, and manufacturing contributes 11% of GDP [124,125]. Over 70% of GDP is derived from services, finance, IT, and other non-productive industries. Conceptually, the U.S. trade condition has shifted from a pyramid to a hot air balloon over the past 120 years (Figure 5). Carrying and foreign trades dominate the top two-thirds of the balloon. The service and knowledge sectors fill a greater volume of the domestic trade, sustained not by the productive labor of the U.S., but that of other countries. The balloon, conceptually, remains aloft by exogenous energy, the carrying trade, and imported products.

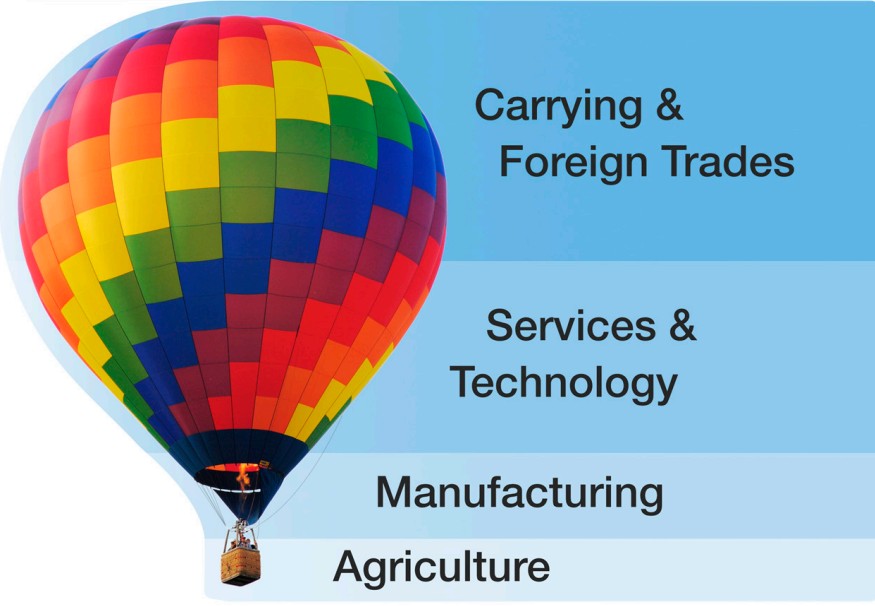

**Figure 5.** Trade balloon, representing the current U.S. economy. Balloon by Pixabay, public access.

*3.4. Interactions between Diversity, Energy, and Trade*

From this interpretive review, it appears diversity, energy, and trade form similar interactions in ecosystems and economies. Highly diverse systems are capable of extracting more energy and material resources from a given space, compared to low-diversity systems. However, transfers of energy and material resources into an economic or ecological space place downward pressure on diversity. In economies, a direct relationship exists between energy and cross-boundary trade. For instance, the use of oil in shipping resulted in a 10-fold increase in the size of cargo ships between 1871 and 1914, compared with cargo ships running coal, a lower energy-dense material [29], or those powered by wind. In turn, goods could be shipped more cheaply, increasing foreign trade profits, which were invested into greater scales of production, putting downward pressure on diversity in affected nations and industries.

3.4.1. Financial Resource Transfers, Homogenization, and Economic Productivity

Resource transfers are also facilitated by financial transfers such as loans, bailouts, and subsidies, which tend to perpetuate large homogenized industries. Hence, financial resources are also categorized as exogenous or endogenous, depending on their origin. Exogenous financial transfers can have impacts on the productive capacity and productivity of entire nations. The MOSES economic model of Sweden, for instance, revealed that regular government subsidies in specific industries in the early 1990s propped up large-scale businesses such as shipyards, which were utilizing obsolete equipment and production processes [126]. The result was to sustain low-performing businesses, which may have contributed to Sweden's economic recession in the mid-1990s.

A similar trend is evident in the U.S., where productivity growth has remained relatively flat between 1970 and 2022 [127], a period coinciding with high rates of financial transfers into the U.S. economy, agricultural subsidies of over U.S. $90 billion [128], and government bailouts of large homogenized industries such as pork producers [129], automobile manufacturers, and banks. This 52-year period also experienced increasing U.S. foreign trade and a sharp decline in manufacturing sector diversity. Economists of the '90s hypothesized that out-of-work laborers would get retrained and find jobs in the service sector or tech industry. Instead, a series of unfortunate economic events unfolded, dubbed the China Shock [130]. This included wage decline, drug and alcohol abuse, deflated housing prices, and lower tax revenues for municipalities. The impacts on unemployment and wage growth stem in part from the fact that the "economic footprint" of manufacturing is high, with each employee generating about seven indirect jobs [131]. For perspective, each service sector creates about 1.1 indirect jobs for each direct service job.

Over this same period, exporting nations such as Asia, Central America, and Sri Lanka homogenized their industries, amassing several risks. Their risks include the following: increased costs of labor, land, and other factors of production, which degrades their comparative advantage globally; low-cost exporters run the risk of tariffs, subsidies, or other counter tactics waged by importing countries to protect their industries; or global market contractions occur, rendering export-based industries "overbuilt" for their domestic markets. A combination of these risks struck Sri Lanka between 2019 and 2022, including MAS Holdings, their largest apparel manufacturer. Following Victoria's Secret's demand that MAS Holdings cut production costs steeply, MAS Holdings' best option was to increase output at their Bangladesh facility, reducing demand for Sri Lankan labor [33]. Similar cycles of industrial homogenization, boom, and bust have repeated across seven centuries of capitalist history [21], including 100 market crashes between 1985 and 2014 [132]. A connection between foreign trade and socio-economic instability was documented as far back as the late Bronze Age collapse [133].

In summary, this hypothesis is provided: an exogenous resource that transfers into or out of an economic space exerts downward pressure on economic diversity, with commensurate decreases in resistance and resilience to disturbance in affected industries and regions.

3.4.2. The Friction between Trade and Energy: The Case of New Belgium Brewing

New Belgium Brewing (NBB) is the seventh-largest brewery in the U.S., exporting tasty libations across the nation. A notable leader in sustainability, NBB has made incredible strides in alternative energy, including on-site photovoltaics, anaerobic digestion to convert organic waste to biogas, and novel energy efficiency programs [134]. As a result, their plant in Fort Collins, CO, meets 12% of its electricity demands with on-site generation. The remaining electricity needs are supplied by wind farms in Wyoming, U.S.A. [135], beyond state lines. Despite great efforts, about 88% of NBB energy consumption remains exogenous.

The NBB energy challenge is faced by countless manufacturers scaled to meet national or global consumer needs. They consume a quantity of energy that cannot be met with on-site—or even regional—endogenous energy sources. To survive, large-scale manufacturers must concentrate energy from sources far beyond their production facilities, so that water, glass, hops, and other inputs can be converted into export products. Certainly, some large-scale production facilities can produce a unit of beer, or other products, with less energy than a small-scale producer. However, such efficiencies are often outweighed by the energy required to ship a bottle of beer in refrigerated trucks from Fort Collins to, say, San Francisco. A socio-economic solution to the NBB case is offered in Giordanengo [42].

*3.5. A Systems-Level Hypothesis*

In summary, this systems-level hypothesis is provided: diversity, energy, and trade constitute the foundational components of an economy that, when interacting freely at a regional scale, confer a relatively stable state to an economy (i.e., relatively high resilience and resistance to disturbance). As energy and material resource transfers into or out of a regional system increase, multiple forms of diversity decline, conferring lower stability to the economy. The threshold of material and energy transfers, beyond which diversity declines, varies depending on one's SEI context.

*3.6. The Basic Operational Unit of the Global Market Economy*

Understanding an economy's foundational components, and their interactions, requires the appropriate boundary condition (i.e., the *basic operational unit*) to be defined. The appropriate boundary condition also depends on the complexity of the system in question. Basic operational units include organ systems, individuals (e.g., Adam Smith), and populations (e.g., the people of Scotland). Each shire in Scotland is also a basic operational unit, as is the Scottish economy, nested in the U.K. The basic operational unit of the global market economy is defined herein as an economic region, similar to the concept of ecoregions.

In nature, two adjacent ecoregions persist as basic operational units (Figure 6). Though wind and wildlife transmit pollen and seeds across ecological boundaries, and animals migrate freely across them, resulting in the exchange of genetic information, the species composition and structure of each forest remain distinct. Resource transfers within each ecoregion are greater than resource transfers between ecoregions.

Restoration ecologists recognize that ecosystems cannot persist at very small scales (e.g., one acre of prairie). Similarly, most economies cannot persist at scales as small as a single city, and maintain their self-regulating qualities. There are ecological and economic reasons for this. In a restored tallgrass prairie, for instance, the disturbance patterns and plant–animal interactions required to maintain a self-regulating ecosystem occur at scales larger than one acre. There is also a relationship between area and diversity, as richness increases in correlation with area [86], and then levels off as the sampling area increases [136]: the species–area curve.

In economics, besides the diversity factors, theories such as *comparative advantage* [137] also inform why a modern economy must operate at a scale greater than a city [42]. When an economy is large enough (e.g., several cities and counties), it can harbor multiple businesses, competing and collaborating within the economic space to gain efficiencies via unique (i.e., context-specific) compositions of skills, materials, and technology.

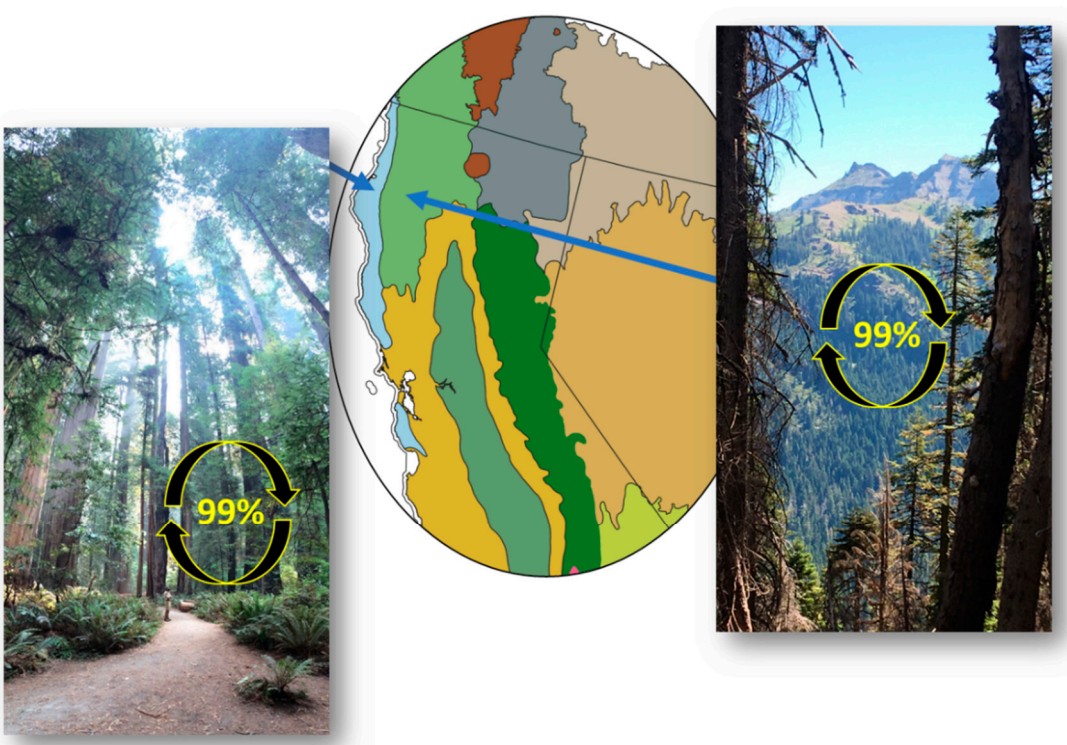

**Figure 6.** Adjacent level III ecoregions in NW California (U.S.A.): Coastal Range (redwood forest) and Klamath Mountains (mixed evergreen forest). The quantity of resource transfers within each system is far greater than the transfers between systems. Spatial data from Omernik [138].

For example, a high diversity of U.S. paper companies that existed in the Midwest in the 1800s developed means to convert recycled rags and straw to produce low-cost newsprint for regional and national consumers while yielding resource-use efficiencies [139]. As U.S. policies—Bretton Woods Agreement, Washington Consensus, etc.—shifted in favor of imported paper and other wood products from different SEI contexts (e.g., Canada and Mexico) in the 20th century, hundreds of U.S. wood and paper mill extinctions occurred. Today, the U.S. paper industry struggles to survive [139], in many cases requiring subsidies to exist. This is not to say the theory of comparative advantage is invalid. Rather, the theory may be most valid, and minimize exploitative tendencies, when applied within the correct boundary condition: regional scales, not global scales.

Another example comes from Salinas de Guaranda in the Andes mountains of Ecuador, where salt mining dominated the economy through the 1960s, primarily serving export markets; economic diversity was low. In the 1970s, Salinas de Guaranda began strengthening trade ties among twenty-four communities encompassing multiple provinces, raising their productive capacity. Today, cooperation, competition, and comparative advantage all operate in this region, which produces over 250 "Salinerito" products for only regional and national markets [41].

## 4. Conclusions

Many nations, regions, and cities dependent on the global marketplace can support high levels of consumption by importing relatively cheap products from different SEI contexts (trade component), facilitated by large quantities of low-cost fossil fuels (energy component). Such a model can generate financial efficiencies, but at the expense of resource-use efficiencies (e.g., waste heat and material waste), and the degradation of other system components (e.g., diversity, social conditions, or soil fertility). This can diminish a nation's or region's NDP and productive capacity, while causing an imbalance of foundational components.

The more unbalanced an economy's structure, the more amplified are the external forces acting on it over time, requiring more maintenance. This translates to more capital being allocated to combat graft, corruption, social unrest, environmental degradation, climate change, etc. That same capital is unavailable to sustain or restore an economy's productive capacity and net domestic productivity, or to provide other social, environmental, or economic benefits. Such an economy is poorly functioning, less capable of sustaining the needs of its chief beneficiaries, humans, while threatening their chief benefactor, nature. Restoring our economies according to the rules of self-regulating ecosystems has significant implications for green infrastructure, circularity, economic restoration, and other sustainable efforts.

### 4.1. Implications for Green Infrastructure

A common goal of green infrastructure is to protect natural resources, and provide various environmental, economic, and social benefits. This includes ecological restoration [140]. Given the interwoven nature of the earth's ecological and economic systems, and of its rural and urban capital, green infrastructure must also include economic restoration: restoring the balance of diversity, energy, and trade in regional economies. Two examples follow. In forested regions of the world, selective logging is used to provide jobs and industrial materials while improving water quality, contributing to carbon neutrality, reducing the risk of catastrophic wildfires, etc. In Bellemont, Arizona (U.S.A.), the Restoration Forest Products mill generates high-value dimensional timber (Figure 7) while yielding multiple byproducts: two forms of biomass fuel, increased understory productivity for grazing animals and pollinators, animal bedding, etc. The efficiencies gained contribute to the region's NDP and productive capacity, and integrate rural and urban capital. Critical for such green infrastructure solutions, the Bellmont mill is profitable enough to attract ample investment [58].

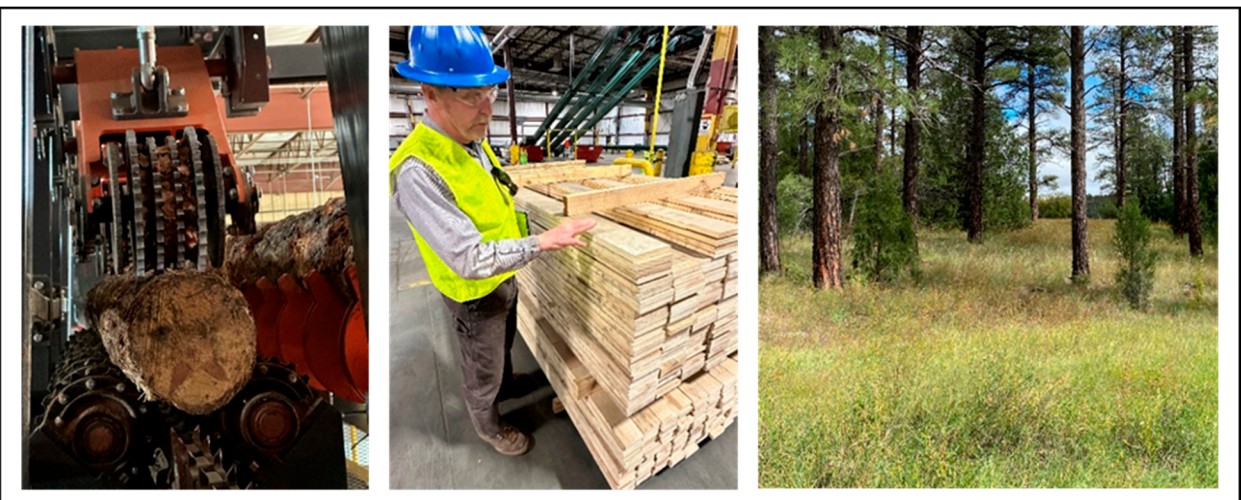

**Figure 7.** (**Left**): Restoration Forest Products mill in Bellemont, AZ. (**Center**): Primary revenue provided by engineered wood, with secondary revenue from several byproducts. (**Right**): Understory of thinned forest, increasing in blue grama (*Bouteloua gracilis*), a native grass with high protein, critical for grazing animals in late summer.

On the contrary, in states like Colorado, felled trees from thinning projects are rarely used for timber or to heat homes. Instead, logs are typically chipped on site or left in piles to burn. Since the trade liberalization policies of the '80s, decades of low-cost timber imports from Canada and elsewhere, coupled with cheap fossil fuels to heat and power buildings, have rendered Colorado's timber industry unviable, with very low diversity.

A second example comes from Sri Lanka, where a steep imbalance of payments began building in the 1980s, associated with exorbitant debt, increased reliance on imports, and other factors [98,141]. These imports included synthetic fertilizers, herbicides, and pesticides. Following Sri Lanka's 2019 economic implosion, coupled with a spike in the cost of imported farm inputs, the nation's export-based agricultural system crashed, alongside textiles and other industries. Farmers stormed the presidential palace in July 2022, sparking a global media frenzy pointing to President Rajapaksa's organic agriculture mandate as the root of the problem; the root, however, goes deeper.

Just as we gain insights into ecosystem structure and functions by observing the system's recovery following disturbance, I visited Sri Lanka in 2023 to interview organic farmers, business owners, scientists, and economists, to observe economic recovery following collapse. Sri Lankan farmers today create nitrogen from six endogenous sources, including four on-farm plant species. Endogenous fertilizers sustain not only tea plantations and subsistence farmers [142], but also profitable vegetable markets [143]. Even coconut farmers such as Gerard Gamini produce a wide range of products for regional and global markets using endogenous chicken manure, far cheaper in 2023 than exogenous fertilizers [144].

Structural diversity and richness in Sri Lankan agriculture are showcased by Ahinsa Tea, which grows over 25 agronomic species on one shade-grown plantation [142]. The plantation is located within the habitat buffer area of Sinharaja Forest Preserve, an IUCN biodiversity hotspot. A key goal of Ahinsa Tea, according to its founder, Dr. Abeygunawardana, is to increase sustainable economic use within the buffer, by developing biodiverse organic tea plantations that sustain local farm families.

Like most economies, the Sri Lankan economy requires significant restoration efforts to attain a self-regulating state, due to their trade and energy imbalances. A critical input to the economy remains imported fossil fuels, totaling U.S. $5 billion in 2022, a GDP deduction. The quantity of potential renewable energy sources (e.g., biofuels, solar, wind, hydroelectric) in Sri Lanka is estimated to exceed the nation's energy consumption [99]. However, some believe that the finances needed to develop this endogenous energy potential would require Sri Lanka to increase domestic trade while minimizing foreign trade, without amassing substantial foreign debt [96,143].

*4.2. Ecological Succession: Implications for Economic Restoration*

Succession, a key process that informs ecological restoration designs, has been illuminated by over a century of research [54,145–150]. *Ecological succession* is the change in plant community composition and structure over time, in response to disturbance. Disturbances such as a severe wildfire can destroy every tree in a forest, obliterating ecological niches while transforming topsoil to ash (Figure 8).

Relevant to economics, a relationship exists between productivity, diversity, and succession. During the early successional stages after wildfire, pioneer species such as annual grasses and forbs generate the bulk of productivity. As succession progresses, slower-growing species begin flourishing as the web of diversity becomes interlaced and tightened. Decades pass, and late successional plants begin to define the community, including mature shrubs and trees. Absent another wildfire, a very dense forest may develop. The economic equivalent of a very late successional stage would be an industry dominated by few large businesses. In the late stage, resources remain "locked up", largely unavailable to faster-growing species (or businesses). Inevitably, disturbances such as fire—or a major supply chain disruption—do arise, liberating resources for pioneer entities, and the succession cycle repeats.

Changes in diversity and productivity are not ever increasing over time, with a peak in the late-successional stage. Instead, a peak in biodiversity and net productivity tends to occur in mid-successional stages [52,54,151–153], shown conceptually in Figure 9. This phenomenon is known as the intermediate disturbance hypothesis (IDH), originally proposed by Connell [75]. In mid-successional stages, community structure and composition are

shaped by smaller disturbances such as blowdowns of individual trees, insect outbreaks, or minor ground fires, liberating resources for other species. The increase in diversity raises the system's capacity to capture resources, contributing to greater resource-use efficiency, and hence higher net productivity.

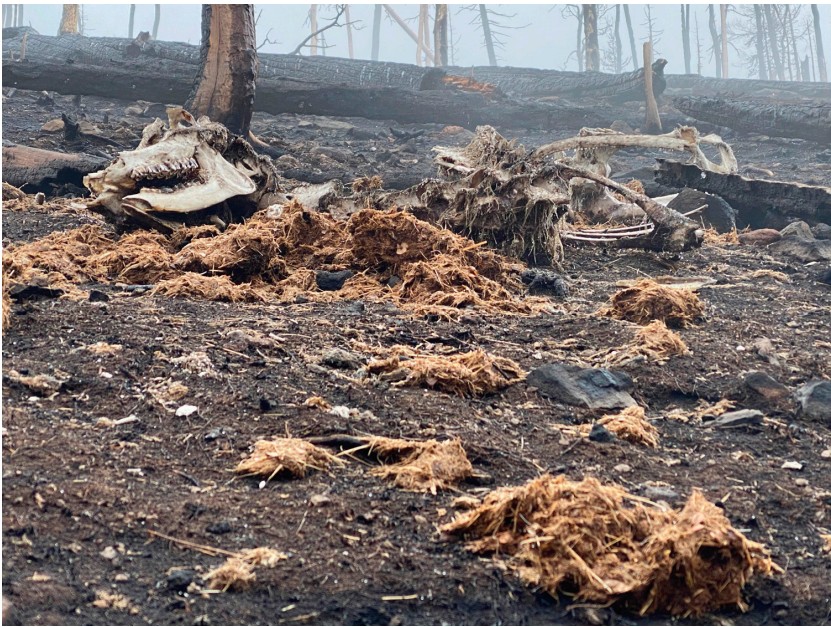

**Figure 8.** Scorched forest and charred elk in the 2020 East Troublesome Fire (U.S.A.), which threatened water supplies for the regionally important Colorado–Big Thompson Project.

Though *economic* succession studies are limited, disturbance–diversity–productivity patterns similar to those observed in ecosystems are evident. In economies, disturbances that shift the system from late- to mid- or early-successional stages drive the process of creative destruction described by Joseph Schumpeter [154]. However, the connection between disturbance, creative destruction, and productivity is indirect. When disturbance strikes (e.g., recessions, industry shake-ups, or disruptive technologies), outdated business structures and processes are replaced by new business entrants and processes, which tend to yield higher rates of innovation [155–157]. Because some of the older business structures do prevail after a disturbance, the addition of new businesses can result in a rise in aggregate diversity. Hence, peak diversity co-occurs with high rates of innovation, which drives productivity gains, a basic requirement for economic growth. Other correlations between diversity and productivity were described earlier.

Countering the process of creative destruction, government programs and policies such as subsidies, bailouts, and loan programs, together with globally-oriented energy and trade policies, tend to homogenize economic landscapes, reinforcing a late successional condition.

In ecology, exceptions to the IDH are summarized by Huston [78] and Hodapp et al. [79]. Relevant to foundational components, Huston noted an ecological exception that has a functional equivalent in economics: when exogenous resources are transferred into the system. Other ecological exceptions include studies that include just one or two taxa (e.g., just upper canopy trees, or just ungulates), or are limited in their spatial or temporal scales [78,158]. Further, ecosystem management decisions such as wildfire suppression can also influence succession processes, producing very late successional communities whose biomass accumulation rates are lower than in previous stages (e.g., slower-growing trees, increased surface accumulation of wood, greater soil organic matter storage, etc.). Lacking adequate disturbance, ecosystems can cross a threshold of biomass (i.e., capital) accumulation, where diversity and net productivity begin declining (Figure 9). These late successional

systems are often more susceptible to more intensive disturbances such as severe wildfires, wide-scale insect outbreaks, etc.

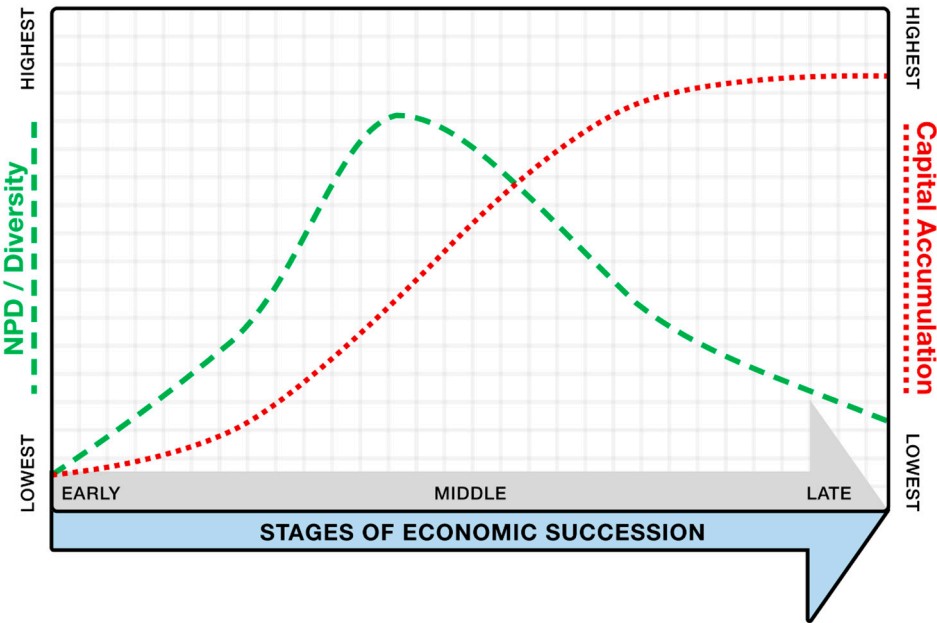

**Figure 9.** Estimates of aboveground net primary production (NPP), net ecosystem production (NEP), and biomass accumulation over the course of ecological succession, after Odum [52] and Gough et al. [151]. NEP can be negative following a disturbance [159].

In many regards, the U.S. economy appears to be exhibiting late-successional characteristics, where per capita GDP growth has dropped from 8% in the 1970s to less than 3% in the twenty-teens [160]. Real wages have remained flat to declining [161–164], wealth has become concentrated into fewer entities [15,17], and productivity returns on investment have declined [165]. If government bailouts, trade policies, monetary policies, and subsidies can shift economic succession to the right, can policies and programs shift succession to the middle, to stimulate diversity and its benefits?

### 4.3. Economic Restoration

Current use of the term economic restoration is restricted to the economic value of ecological restoration (e.g., the restoration economy), or to beautifying an urban area, rather than to restoring economic processes and functions. To clarify, I offer this definition: *Economic restoration* is the process of rebuilding economic diversity, endogenous trade, and endogenous energy systems within a regional economy. Due to this article's length, readers interested in additional data and analysis related to economic restoration, and its social, economic, and environmental implications, may contact the author for a digital copy of "Ecosystems as Models for Restoring Our Economies".

### 4.4. Economic Restoration Policy Implications

Several policy and program (P & P) implications stem from the results above, as related to diversity, energy, trade, and boundary conditions. Three inter-related policy implications are provided below, with others provided in Giordanengo [42]. These P & Ps are relevant not only to governments, but to NGOs, universities, and private entities working in the sustainability, green infrastructure, and circularity spaces. Specific P & Ps will vary by SEI context.

Policy implication 1: Energy security, carbon neutrality, and similar goals are facilitated not only via technology changes, but by increased economic diversity at regional scales. To stimulate diversity, P & Ps must be crafted to increase endogenous energy production and

trade at regional and national scales, while minimizing exogenous energy and foreign trade. Many means exist to accomplish this, including import substitution, flexible manufacturing, cooperatives, ecological clustering, and similar system-level solutions.

Policy implication 2: Fiscal P & Ps that stimulate cross-boundary trade, such as subsidies, bailouts, loan programs, and tax breaks, have historically placed downward pressure on diversity. To facilitate the benefits of diversity described in this paper, fiscal P & Ps should be refocused to stimulate new business entrants in waste-to-resources, endogenous energy, extractive (e.g., agriculture and logging), and manufacturing industries at regional levels.

Policy implication 3: In support of policy implications above, a variety of P & Ps are available to facilitate the restoration of regional economies, such as multi-agency economic compacts, inter-agency trade and fiscal policies, regional energy development programs, and similar mechanisms. These P & Ps do not require the redrawing of jurisdictional boundaries (e.g., city, county, or state), or the creation of new currencies. Rather, regional economic zones can be established on watershed boundaries, ecological boundaries, or similar boundaries, which may cross multiple existing jurisdictions.

**Funding:** This research received no external funding.

**Data Availability Statement:** Data, tables, and additional text available at www.economicrestoration.org (accessed on 28 June 2023).

**Conflicts of Interest:** The author declares no conflict of interest.

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
