# Peer review of "The Foundational Components of Self-Regulating (Sustainable) Economies and Ecosystems: Implications for Green Infrastructure and Economic Restoration"

_land, doi:10.3390/land12112044_

Round 1
Reviewer 1 Report
Comments and Suggestions for Authors
It is of great significance to study the foundational components of self-regulating economies and ecosystems. The manuscript needs to be improved in the following two aspects.
1.Why the three foundational components of both economies and ecosystems structure are diversity, energy and trade, and on what basis ? It needs to strengthen the argument.
2.The specific interaction and mutual influence mechanism among diversity, energy and trade in economies and ecosystems are suggested to be shown by figures.
Author Response
Thank you kindly for your insightful comments, which I will address in the final manuscript to the extent possible. Below are a few specific responses.
REVIEWER: Why the three foundational components of both economies and ecosystems structure are diversity, energy and trade, and on what basis ? It needs to strengthen the argument.
AUTHOR RESPONSE: I have made revisions to address this point, and hope the explanation provides additional clarity to the reader.
REVIEWER: The specific interaction and mutual influence mechanism among diversity, energy and trade in economies and ecosystems are suggested to be shown by figures.
AUTHOR RESPONSE: Excellent idea, which I have developed in draft form. If time exists prior to publication, I will formalize my draft figure to include in the final manuscript.
Reviewer 2 Report
Comments and Suggestions for Authors
The problem of the article is relevant and current, in the context of global economies and the dynamics of development and its sustainability.
The research is extensive and includes historical analyses of the economic behaviors of nations, of the dominant economic system by epoch, and of the respective paradigms.
It should be noted that the introduction requires an approach of greater description of the problem under study and its economic, social and environmental repercussions, requires a definition of the objectives (in a clear and structuring manner of the research presented) and of the way in which the article is organized/structured.
It presents problems of citations throughout the text that should be rectified, as well as promoting the formatting of graphs and figures according to the rules of the Journal.
It is a descriptive text and the methodologies followed, along with the approaches and analysis produced, are more adequate for a book chapter. This study requires significant improvements in organization, definition of objectives, and analysis processes that must go beyond theoretical approaches to the verification and testing of hypotheses.
Comments on the Quality of English LanguageModerate editing of English language required
Author Response
Thank you kindly for your insightful comments. I am revising the manuscript now, and hope the changes adequately reflect your points. Below are some specific responses.
REVIEWER: It should be noted that the introduction requires an approach of greater description of the problem under study and its economic, social and environmental repercussions, requires a definition of the objectives (in a clear and structuring manner of the research presented) and of the way in which the article is organized/structured.
REVIEWER: It presents problems of citations throughout the text that should be rectified, as well as promoting the formatting of graphs and figures according to the rules of the Journal.
AUTHOR RESPONSE: I will work on rectifying these with the editorial staff.
REVIEWER: It is a descriptive text and the methodologies followed, along with the approaches and analysis produced, are more adequate for a book chapter.
AUTHOR RESPONSE: I have made some revisions to address this concern.
REVIEWER: This study requires significant improvements in organization, definition of objectives, and analysis processes that must go beyond theoretical approaches to the verification and testing of hypotheses.
AUTHOR RESPONSE: If I understand your comment correctly, I have addressed this in the methods section by clarifying that the systems approach I used does not adhere well to traditional hypothesis testing. Instead, I drew from a cross-cutting approach, which I clarified further in the subsequent draft. I have also added and clarified the objectives. As other reviewers were in support of the original hypotheses, objectives, and format, I am working to strike a balance of those reviewers input while also incorporating what I believe your suggestions are.
Reviewer 3 Report
Comments and Suggestions for Authors
A very good paper see attached some suggestions.

English is excellent and the presentation well considered.
Author Response
Hello,
Thank you kindly for your thoughtful comments to the draft manuscript, which I will incorporate into the final. Below are some specific responses.
REVIEWER: The inquisitive reader will want to know how different elements of global investment – such as carbon free energy sources and the move from fossil fuel energy to environmentally sustainable energy such as wind, solar and nuclear may produce dividends.
AUTHOR RESPONSE: Great point. I will include a summary in the added policy section.
REVIEWER: The politics of the day and the operation of political choices needs to be considered. In fact much of what is said in the paper makes a good read for political policy makers. There is also no mention of NGOs or lobby groups. Both are part of civil society and their existence needs to be factored into the way economic choices are made and the reason behind political decisions.
AUTHOR RESPONSE: Good point, which I will address in the policy section as space allows.
REVIEWER: It would be useful to have some discussion addressing how the author’s analysis is similar to other economists or other academics. This would provide a useful point of comparison between each thinker and writer.
AUTHOR RESPONSE: I believe I can provide some detail in the final, though space is limited.
REVIEWER: It would also be helpful to set out (briefly) some ideas on the policy implications for policy makers and politicians. This is an important aspect of the conclusions reached in the paper.
AUTHOR RESPONSE: Very important suggestion. Space is limited, but I will provide a few brief policy recommendations.
REVIEWER: The paper might be shortened in places without losing its substantial force or standing. Some sections might be shortened at the start of the paper.
AUTHOR RESPONSE: Agreed, but I am not sure if I will be able to provide "shortening" edits before final submission.
NOTE: From your comments, I am curious to know your affiliation, as we may have some mutual interests that would be fruitful to explore. If you are comfortable, feel free to contact me at john@economicrestoration.org.
Round 2
Reviewer 2 Report
Comments and Suggestions for Authors
The article continues to reveal an exaggerated extension for a scientific article, which by its form of approach, structure, descriptive exploration of variables and socio-economic contexts should be rethought and reorganized.
There have been some improvements in the objectives and methodological aspects, some subtitles have been revised, however there are still deep problems of organization and formatting in particular with the citations in the text. I believe that the proposed tutorial should be revised and simplified, allowing a clearer analysis of the text. I believe that the complexity and size of the article and its organization creates problems in defining the title itself. It is a 9-page introduction, which is incomprehensible, relating various conceptual approaches and frameworks inappropriate for an introduction. Do not hesitate a state of the art on the problem (literature review), with its own organization and autonomous from the introduction. The results are developed over 13 pages, mixing in a poorly structured way analyses of variables, descriptions of concepts and analyses of contexts that do not contribute with value and rigor to a scientific work of this nature. It must be thoroughly revised, reorganized, synthesized and improved in order to be accepted.
Background work is requested to reorganize the article into a simpler and more objective structure. Some parts of the text, especially descriptive and (socio-economic) equdramento could be systematized in conceptual tables.It should be rethought as an article and not as a book chapter (although as it is developed it fits into a book chapter). A more detailed and systematized analysis of the results and if possible with supporting infographics would be desirable.
Comments on the Quality of English LanguageModerate editing of English language required
Author Response
Reviewer: The article continues to reveal an exaggerated extension for a scientific article, which by its form of approach, structure, descriptive exploration of variables and socio-economic contexts should be rethought and reorganized.
Response: Understood. In retrospect, you are correct (if I'm reading between the lines), that a systems level analysis is not appropriate for a standard scientific process. Or at least I have not seen a standard scientific process address this subject matter well. I will be resubmitting this as a review rather than an article, so thank you for this recommendation. It's a good one.
Reviewer: There have been some improvements in the objectives and methodological aspects, some subtitles have been revised, however there are still deep problems of organization and formatting in particular with the citations in the text. I believe that the proposed tutorial should be revised and simplified, allowing a clearer analysis of the text. I believe that the complexity and size of the article and its organization creates problems in defining the title itself. It is a 9-page introduction, which is incomprehensible, relating various conceptual approaches and frameworks inappropriate for an introduction. Do not hesitate a state of the art on the problem (literature review), with its own organization and autonomous from the introduction. The results are developed over 13 pages, mixing in a poorly structured way analyses of variables, descriptions of concepts and analyses of contexts that do not contribute with value and rigor to a scientific work of this nature. It must be thoroughly revised, reorganized, synthesized and improved in order to be accepted.
Author: I will dramatically reduce the introduction, and restructure article in next draft. I am not sure what the specific issues are with the citations, so I will work with MDPI editors on that. I will reduce extent of results and try to streamline.
Reviewer: Background work is requested to reorganize the article into a simpler and more objective structure. Some parts of the text, especially descriptive and (socio-economic) enqudramento could be systematized in conceptual tables. It should be rethought as an article and not as a book chapter (although as it is developed it fits into a book chapter). A more detailed and systematized analysis of the results and if possible with supporting infographics would be desirable.
Author: I will consider additional results and possibly some infographics.